

# Decoders configurations based on Unet family and feature pyramid network for COVID-19 segmentation on CT images

Hai Thanh Nguyen[1], Toan Bao Tran[2,3], Huong Hoang Luong[4] and Tuan Khoi Nguyen Huynh[4]

[1] College of Information and Communication Technology, Can Tho University, Can Tho, Vietnam
[2] Center of Software Engineering, Duy Tan University, Da Nang, Vietnam
[3] Institute of Research and Development, Duy Tan University, Da Nang, Vietnam
[4] FPT University, Can Tho, Vietnam

## ABSTRACT

Coronavirus Disease 2019 (COVID-19) pandemic has been ferociously destroying global health and economics. According to World Health Organisation (WHO), until May 2021, more than one hundred million infected cases and 3.2 million deaths have been reported in over 200 countries. Unfortunately, the numbers are still on the rise. Therefore, scientists are making a significant effort in researching accurate, efficient diagnoses. Several studies advocating artificial intelligence proposed COVID diagnosis methods on lung images with high accuracy. Furthermore, some affected areas in the lung images can be detected accurately by segmentation methods. This work has considered state-of-the-art Convolutional Neural Network architectures, combined with the Unet family and Feature Pyramid Network (FPN) for COVID segmentation tasks on Computed Tomography (CT) scanner samples from the Italian Society of Medical and Interventional Radiology dataset. The experiments show that the decoder-based Unet family has reached the best (a mean Intersection Over Union (mIoU) of 0.9234, 0.9032 in dice score, and a recall of 0.9349) with a combination between SE ResNeXt and Unet++. The decoder with the Unet family obtained better COVID segmentation performance in comparison with Feature Pyramid Network. Furthermore, the proposed method outperforms recent segmentation state-of-the-art approaches such as the SegNet-based network, ADID-UNET, and A-SegNet + FTL. Therefore, it is expected to provide good segmentation visualizations of medical images.

# INTRODUCTION

Since the end of 2019, the COVID-19 pandemic has affected human lives heavily. In addition to a considerable number of deaths, the economic losses caused by the COVID-19 pandemic are hard to estimate. Based on the report published by the World Bank, the world economy is estimated to decline by 4.3% in 2020 (https://www.worldbank.org/en/news/press-release/2021/01/05/global-economy-to-expand-by-4-percent-in-2021-

Corresponding author
Hai Thanh Nguyen,
nthai.cit@ctu.edu.vn

vaccine-deployment-and-investment-key-to-sustaining-the-recovery). Many observers commented that this significant decline only occurred during the Great Depression and two world wars. Since a pandemic is so damaging to health and society, it is worthwhile to gather great resources to fight it. According to experts, to combat the COVID-19 epidemic, the COVID-19 vaccine is the best solution. COVID-19 has also infected over 157 million cases and caused more than 3.2 million deaths according to World Health Organisation (WHO) (https://www.who.int/emergencies/diseases/novel-coronavirus-2019) across 223 countries, areas, or territories with cases. The number of deaths still increases thousands every day. Therefore, COVID-19 diagnosis improvement is essential not only for treatment but also for preventing its infection. Fortunately, we are in an era where science and technology have been walking a decisive step forward with tremendous achievements. This development has changed the world, and it also promotes and speeds up the enhancement of medicine and other fields.

Recent medical imaging analysis studies have revealed huge benefits with the current advancements in technology and have been considered an essential role in determining patients' condition and status. Achievements information technologies have greatly supported doctors during the diagnosis and treatment of patients. Also, there is a global trend in applying information technology to health care. Hospital information systems, clinical decision-making support systems, telemedicine, virtual reality, and health information highway have been developing and appearing globally so that public health care gets better and better. The government in various regions has developed national health information technology programs to computerize and digitize health records. Many research and application programs are implemented in hospitals and health facilities with hospital information systems (*Ferdousi et al., 2020*), communication systems (*Nayak & Patgiri, 2021*; *Belasen et al., 2020*), robot-based surgeon systems (*Lee et al., 2021*), and nursing care information systems (*Booth et al., 2021*). Medical records, images of x-ray, ultrasound, magnetic resonance imaging, positron-emission tomography become rich and diverse.

Modern medicine with information technology applications can make disease diagnosis faster based on various clinical symptoms and subclinical symptoms (subclinical diagnosis). In subclinical diagnostics cases, doctors usually evaluate and examine images generated and screened from medical imaging devices and equipment. Modern and high-tech medical machines with computer support software make the image clearer and more accurate with a very high resolution. The diagnostic imaging methods are diversified, such as radiological diagnosis, ultrasound imaging, ultrasound-color Doppler, endoscopic images (commonly used as gastrointestinal endoscopy and urinary endoscopy), Computed Tomography (CT) Scanner, Magnetic Resonance Imaging (MRI) and so on.

Image segmentation is to divide a digital image into various parts, which can be the collections of pixels or superpixels (*Shapiro, 2001*). The goal of image segmentation aims to simplify and or represent an image into something more meaningful and easier to analyze. In recent years, deep learning algorithms have provided great tools for medical segmentation, which plays an essential role in disease diagnosis and is one of the most

crucial tasks in medical image processing and analysis. Diagnostic based on segmented medical images holds an essential contribution to improving accuracy, timeliness, and disease diagnosis efficiency (*Saood & Hatem, 2021*; *Budak et al., 2021*; *Zhou, Canu & Ruan, 2020*; *Raj et al., 2021*; *Yakubovskiy, 2020*). For example, for ultrasound images, the physician and doctors can accurately detect and measure the size of the solid organs in the abdomen and abnormal masses on the segmented areas in the images (*Ouahabi & Taleb-Ahmed, 2021*; *Zhou et al., 2021*). Another type of medical image is Chest x-ray (*Rahman et al., 2020*), where cancer tumors can be detected and segmented for surgeons and efficient treatment monitors. The CT scanner images marked abnormal regions also help physicians identify signs of brain diseases, especially identifying intracranial hematoma, brain tumors (*Ramesh, Sasikala & Paramanandham, 2021*; *Munir, Frezza & Rizzi, 2020*). Signs of the disease can be revealed *via* such segmented images, but sometimes these signs can be too small to be observed by humans. Moreover, in a short time, doctors may have many patients for diagnosis simultaneously. Besides, it takes so much time to train a doctor to perform medical image analysis. Artificial intelligence algorithms can outperform human ability in image classification and provide techniques to interpret the results (*Geirhos et al., 2018*). Therefore, leveraging artificial intelligence's development with segmentation image techniques is crucial to accelerate medicine advancement and improve human health. With an outbreak of the COVID-19 pandemic, it requires significant efforts from all citizens worldwide to stop the pandemic, but human resources seem insufficient. Technology-based medical approaches are necessary and urgent to support humans to reduce and prevent the pandemic, so algorithms on image processing for diagnosis of COVID-19 have attracted the attention of numerous scientists. The efficiency of the Unet family in image segmentation and outstanding image classification performance of well-known convolutional neural network architectures revealed in a vast of previous studies has brought a significant research question on the benefits of their combinations to enhance the performance in COVID-19 lung CT image segmentation. This study has leveraged well-known deep learning architectures as encoders and Unet family, Feature Pyramid Network techniques as decoders to produce segmentation on chest slices for supporting COVID-19 diagnosis. The principal contributions include as follows:

- Numerous configurations generated by combining five well-known deep learning architectures (ResNet, ResNeSt, SE ResNeXt, Res2Net, and EfficientNet-B0) and Unet family are evaluated and compared to the state-of-the-art to reveal the efficiency in the COVID-19 lung CT image segmentation. Moreover, we also include Feature Pyramid Network (FPN), a famous architecture for segmentation tasks in configurations for comparison.
- The visualizations including segmented regions in lungs are examined with various metric performances and exhibit similar infected areas compared to the ground truths.
- Augmentation on the COVID-19 lung images dataset is performed with mirror, contrast, and brightness transforms. In addition, gamma and Gaussian noise are manipulated before adding spatial transforms. Such techniques help to enhance the

segmentation performance. Besides, we evaluate the performance of COVID-19 segmentation without augmentation techniques.

- The training time and inference time are measured and compared among the considered configurations.
- From the obtained results, we found that the integration between SE ResNeXt and Unet ++ has revealed the best performance in COVID-19 segmentation tasks.

The rest of this study is organized as follows. 'Related Work' introduces the main related works. 'Method' presents a brief introduction of the segmentation network based on encoder-decoder architectures, the dedicated loss function for segmentation tasks, and augmentation techniques. Afterward, we present our settings, the public COVID-19 dataset, and the evaluation metrics in 'Evaluation'. 'Experimental Results' exhibits and analyzes the obtained results. We conclude the study and discuss future work in 'Conclusion'.

## RELATED WORK

Machine learning for medical imaging analysis has gained popularity in recent years. Advancements in computer techniques have also been proposed with an increase in quality and quantity. To support doctors better, researchers have focused on model explanation and segmentation methods for medical images. 'Applications of deep learning in healthcare' examines the robust studies of deep learning in the healthcare sector. 'Applications of deep learning in COVID-19 detection' reviews the main related approaches in the domain of COVID-19 detection from CT images.

### Applications of deep learning in healthcare

*Ravi et al. (2017)* presented several robust applications of deep learning to health informatics. We have obtained benefits from rapid improvements in computational power, fast data storage, and parallelization, and so there are more and more efficient proposed models for health services. Furthermore, *Srivastava et al. (2017)* gave us an overview of recent trends and future directions in health informatics using deep learning. In another study, *Huynh et al. (2020)* introduced a shallow convolutional neural network (CNN) architecture with only a few convolution layers to perform the skin lesions classification, but the performance is considerable. The authors conducted the experiments on a dataset including 25,331 samples. For discriminating between melanoma and vascular lesion, the proposed model obtained an accuracy of 0.961 and an Area Under the Curve (AUC) of 0.874. Several studies deal with abnormality bone detection (*Chetoui & Akhloufi, 2020*; *Varma et al., 2019*) also revealed a promising result. For instance, *Chetoui & Akhloufi (2020)* developed a deep learning architecture based on EfficientNet (*Tan & Le, 2019*) to detect referable diabetic and diabetic retinopathy on two public datasets, namely EyePACS and APTOS 2019. The proposed method achieved the highest AUC of 0.990 and 0.998 on EyePACS and APTOS 2019, respectively. Similarly, an approach to detect abnormalities on musculoskeletal images has been proposed by using CNN architecture. The authors collected a massive dataset of 93,455 radiographs.

The obtained AUC is recorded of 0.880, sensitivity and specificity of 0.714 and 0.961, respectively.

To recognize COVID-19 from chest CT images, a deep architecture named COVNet has been proposed by *Li et al. (2020)*. Community-acquired pneumonia and healthy control are utilized in the testing phase and collected from six hospitals. AUC, specificity, and sensitivity report the performance. For detecting COVID-19, the proposed method obtained an AUC of 0.96, a specificity of 0.96, and a sensitivity of 0.90. *Shi et al. (2021)* presented a review on emerging artificial intelligence technologies to support medical specialists. The authors also stated that "image segmentation plays an essential role in COVID-19 applications".

### Applications of deep learning in COVID-19 detection

*Saood & Hatem (2021)* proposed an approach for image tissue classification by leveraging segmentation networks, namely SegNet and U-Net. The purpose of using both models is to distinguish the infected and healthy lung tissue. The networks are trained on 72 and validated on ten images. The proposed method obtained an accuracy of 0.95 with SegNet and 0.91 with U-Net. Empirically, the authors also stated that the mini-batch size affects the performance negatively. *Budak et al. (2021)* presented a new procedure for automatic segmentation of COVID-19 in CT images using SegNet and attention gate mechanism. A dataset with 473 CT images has been utilized as the evaluation data. The performance of the proposed method is judged based on Dice, Tversky, and focal Tversky loss functions. The authors reported that the obtained sensitivity, specificity, and dice scores are 92.73%, 99.51%, and 89.61%, respectively.

*Zhou, Canu & Ruan (2020)* proposed an effective model to segment COVID-19 from CT images. In comparison to other existing studies (*Budak et al., 2021*), the model obtained comparable results. For each CT slice, the proposed method takes 0.29 s to generate the segmented results and obtained a Dice of 83.1%, Hausdorff of 18.8. However, the method is conceived to segment the single class and on a small dataset. A recent approach (*Raj et al., 2021*) leverages a depth network, namely ADID-UNET, to enhance the COVID-19 segmentation performance on CT images. The proposed method is evaluated on public datasets and achieved a 97.01% accuracy, a precision of 87.76%, and an $F_1$ score of 82.00%.

## METHOD

This section contains five parts. 'Segmentation network architecture' describes systematically the complete architecture of segmentation models. We present the explanation of the encoders and decoder for a general segmentation architecture are presented in 'Efficient deep learning model-based encoder' and 'Decoders for segmentation network' respectively. Afterwards, the description of loss function and several data augmentation methods are explained in 'Loss function for segmentation task' and 'Data augmentation' respectively.
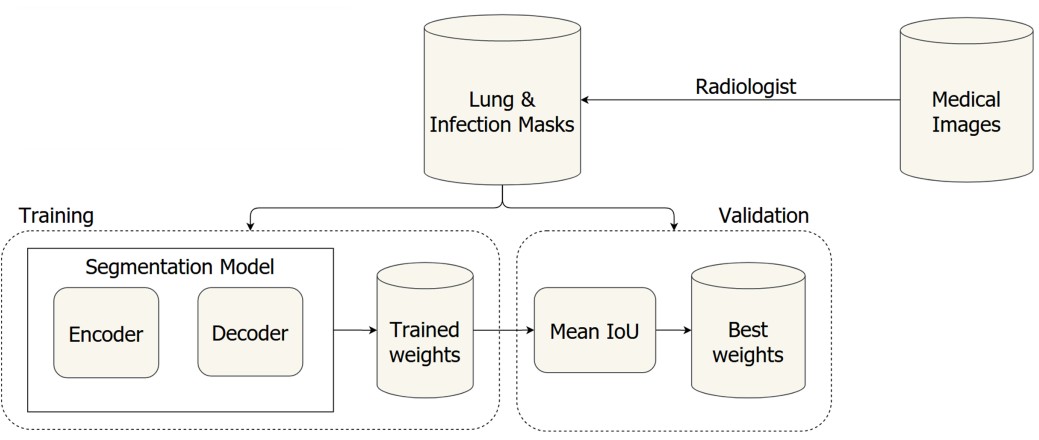

**Figure 1** The proposed architecture of COVID-19 segmentation system on medical images.

**Table 1 The model structures with Unet family and FPN decoder.**

| ResNet 34 | ResNeSt | SE ResNeXt | Res2Net | EfficientNet-B0 |
|---|---|---|---|---|
| Input Layer | Input Layer | Input Layer | Input Layer | Input Layer |
| Output: 1 × 512 × 512 | Output: 1 × 512 × 512 | Output: 1 × 512 × 512 | Output: 1 × 512 × 512 | Output: 1 × 512 × 512 |
| Layer 1 | Layer 1 | Layer 1 | Layer 1 | Layer 1 |
| Output: 64 × 256 × 256 | Output: 64 × 256 × 256 | Output: 64 × 256 × 256 | Output: 64 × 256 × 256 | Output: 32 × 256 × 256 |
| Layer 2 | Layer 2 | Layer 2 | Layer 2 | Layer 2 |
| Output: 64 × 128 × 128 | Output: 256 × 128 × 128 | Output: 256 × 128 × 128 | Output: 256 × 128 × 128 | Output: 24 × 128 × 128 |
| Layer 3 | Layer 3 | Layer 3 | Layer 3 | Layer 3 |
| Output: 128 × 64 × 64 | Output: 512 × 64 × 64 | Output: 512 × 64 × 64 | Output: 512 × 64 × 64 | Output: 40 × 64 × 64 |
| Layer 4 | Layer 4 | Layer 4 | Layer 4 | Layer 4 |
| Output: 256 × 32 × 32 | Output: 1024 × 32 × 32 | Output: 1024 × 32 × 32 | Output: 1024 × 32 × 32 | Output: 112 × 32 × 32 |
| Layer 5 | Layer 5 | Layer 5 | Layer 5 | Layer 5 |
| Output: 512 × 16 × 16 | Output: 2048 × 16 × 16 | Output: 2048 × 16 × 16 | Output: 2048 × 16 × 16 | Output: 320 × 16 × 16 |
| Decoder Layer | Decoder Layer | Decoder Layer | Decoder Layer | Decoder Layer |
| Output: 16 × 512 × 512 | Output: 16 × 512 × 512 | Output: 16 × 512 × 512 | Output: 16 × 512 × 512 | Output: 16 × 512 × 512 |
| Segmentation Layer | Segmentation Layer | Segmentation Layer | Segmentation Layer | Segmentation Layer |
| Output: 3 × 512 × 512 | Output: 3 × 512 × 512 | Output: 3 × 512 × 512 | Output: 3 × 512 × 512 | Output: 3 × 512 × 512 |

## Segmentation network architecture

The overall system architecture is visualized in Fig. 1. We leveraged the segmentation model with architecture to discriminate the COVID-19 infections from the lung on the medical images, including the encoder and decoder. The trained weights are validated on a separated dataset to find the optimal weights during the training section. To segment lung and COVID-19 infection regions from medical images, we utilized the encoder built based on ResNet, ResNeSt, SE ResNeXt, and Res2Net, with the structures are presented in Table 1. Furthermore, the encoder's generated feature maps are handled by the Unet

**Table 2 The complete architecture and configurations of segmentation models.**

| Configuration name | Encoder | Decoder | # of trainable params | Model size |
|---|---|---|---|---|
| $C_1$ | ResNet | | 24,430,677 | 293.4 MB |
| $C_2$ | ResNeSt | | 24,033,525 | 288.8 MB |
| $C_3$ | SE ResNeXt | Unet | 34,518,277 | 414.8 MB |
| $C_4$ | Res2Net | | 32,657,501 | 392.5 MB |
| $C_5$ | EfficientNet-B0 | | 6,251,473 | 72.2 MB |
| $C_6$ | ResNet | | 13,394,437 | 160.9 MB |
| $C_7$ | ResNeSt | | 13,394,437 | 160.9 MB |
| $C_8$ | SE ResNeXt | Unet2d | 13,394,437 | 160.9 MB |
| $C_9$ | Res2Net | | 13,394,437 | 160.9 MB |
| $C_{10}$ | EfficientNet-B0 | | 13,394,437 | 160.9 MB |
| $C_{11}$ | ResNet | | 26,072,917 | 313.2 MB |
| $C_{12}$ | ResNeSt | | 40,498,165 | 468.4 MB |
| $C_{13}$ | SE ResNeXt | Unet++ | 50,982,917 | 612.5 MB |
| $C_{14}$ | Res2Net | | 49,122,141 | 590.2 MB |
| $C_{15}$ | EfficientNet-B0 | | 6,569,585 | 76.1 MB |
| $C_{16}$ | ResNet | | 23,149,637 | 270.8 MB |
| $C_{17}$ | ResNeSt | | 17,628,389 | 211.9 MB |
| $C_{18}$ | SE ResNeXt | FPN | 28,113,141 | 337.9 MB |
| $C_{19}$ | Res2Net | | 26,252,365 | 315.6 MB |
| $C_{20}$ | EfficientNet-B0 | | 5,759,425 | 66.3 MB |

family and FPN. Each encoder has a distinct feature map dimension. The input of the decoder needs to be adapted based on this dependence. Moreover, the encoders leverage the Imagenet pre-trained weights to reduce the computation cost during training. The medical images are pass into the input layer with the dimension of $512 \times 512$. It is forwarded through five different layers with complex structures to extract the most meaningful features.

Afterward, the decoders control the critical stages by generating the masks, including the important regions. The final result obtained a $3 \times 512 \times 512$ array, including three masks. The first mask consists of the non-lung nor COVID-19 regions. The second mask includes the lung's pixels, whereas the final mask reveals COVID-19 infection regions' information. Besides, the combination between the encoder and decoder architectures is presented in Table 2. We considered to use 20 experimental configurations, *i.e.*, $C_i = 1, \ldots,$ 20. In this study, we also present the process diagram of the segmentation model in Fig. 2. The input takes medical images where the output generates two masks consisting of lung and infection regions. *Ronneberger, Fischer & Brox (2015)* and *Long, Shelhamer & Darrell (2015)* demonstrated that the improvement of segmentation tasks recently relied more on the encoder-decoder than other architectures. Furthermore, this architecture was inspired by the Convolutional Neural Network (*LeCun et al., 1989*) and added the decoder network, which effectively tackled the pixel-wise prediction.

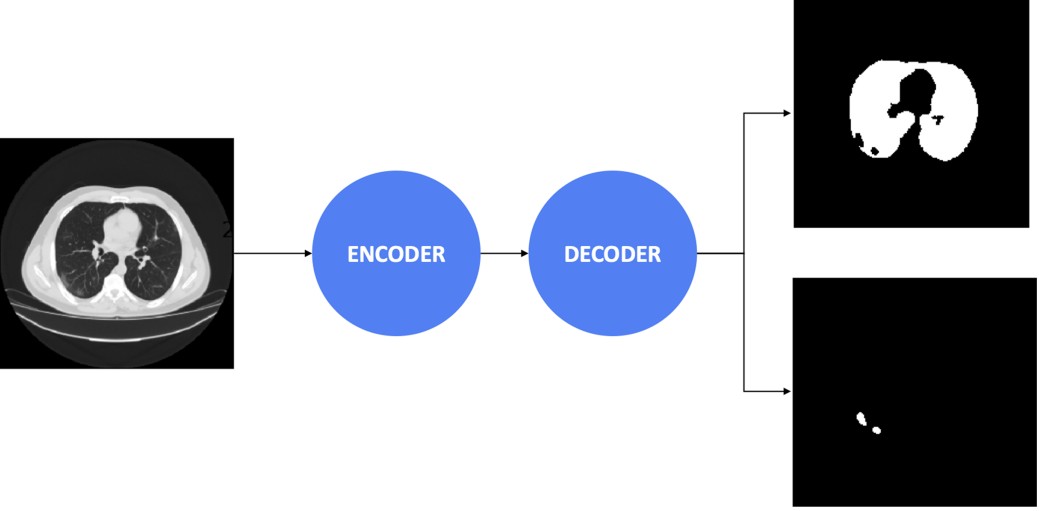

**Figure 2** **The process diagram of segmentation model.**

## Efficient deep learning model-based encoder

In an attempt to improve the performance of image segmentation task on medical images, we leveraged the advantages of numerous pre-trained models as the encoder, which are modern architectures and have impressive performances on classification tasks, such as EfficientNet (*Tan & Le, 2019*), ResNet (*He et al., 2016*), or ResNeSt (*Zhang et al., 2020*). The responsibilities of the encoder are learning features and providing the initial low-resolution representations. The segmentation architecture will refine the encoder's outputs, referred to as the decoder network. To avoid the vanishing gradient problem (*Kolen & Kremer, 2009*) and retrieve the fine-grained information from the previous layer. The skip connections are utilized between the encoder and decoder network or between the encoder and decoder network layers. Assuming $x$ denotes the input, the expected underlying mapping obtaining by training is $f(x)$, the block within the dotted-line box demands to apply the residual mapping $f(x) - x$. In the case of $f(x) = x$, the identity mapping is the desired underlying mapping. The weights and biases of the block within the dotted-line box need to be set at 0.

In the scope of this study, we leveraged ResNet, ResNeSt, SE ResNeXt, and Res2Net as the encoders to generate the features map and transfer the features to the decoder network segmenting the COVID-19 regions on medical images. ResNeSt is a new network inherited from ResNet with an attention mechanism, performed promising results on image classification and segmentation tasks. In this architecture, the feature maps are split into G groups where $G = \#Cardinality \times \#Radix$. The introduction of Cardinality is presented from Resnext (*Xie et al., 2017*), which repeats the bottle-neck blocks and breaks channel information into smaller groups, whereas Radix represents the block of Squeeze-and-Excitation Networks (SENet) (*Hu, Shen & Sun, 2018*). In summary, this architecture combines the Cardinality of Resnext and Attention of Squeeze and Excitation Networks to formulate the Split Attention. In other words, Split Attention is the modification of the gating mechanism. Recently, *Gao et al. (2021)* proposed a novel

building block for CNN constructs hierarchical residual-like connections within one single residual block, namely Res2Net. In other words, the bottle-neck block of the ResNet architecture is re-designed and contributes to increasing the range of receptive fields for each network layer by representing multi-scale features at a granular level. By leveraging kernel size of $7 \times 7$ instead of $3 \times 3$, the computation of multi-scale feature extraction ability is enhanced but achieved a similar cost.

## Decoders for segmentation network

The emergence of artificial intelligence and especially Convolutional Neural Network architectures in computer vision brings the field of image processing to light. Once considered untouchable, several image processing tasks now present promising results like image classification, image recognition, or image segmentation. The image segmentation task's primary purpose is to divide the image into different segment regions, representing the discriminate entity. Compared with classification tasks, segmentation tasks require the feature maps and reconstructing the feature maps' images. In this study, we leveraged the advantages of several CNN-based architectures as the decoder of segmentation models.

### Unet family

We use the implementations of Unet (*Ronneberger, Fischer & Brox, 2015*) with the architecture includes three sections, namely the contraction, the bottle-neck, and the expansion section. Generally, the contraction section is the combination of several contraction blocks. Each block consists of $3 \times 3$ convolution and $2 \times 2$ max-pooling layers. The number of feature maps gets double at each max-pooling layer. It helps the architecture learn complex patterns effectively. Furthermore, the kernels of size $3 \times 3$ are widely used as filters for the widespread deep neural networks (*Chollet, 2017*; *Tan et al., 2019*). Besides, the model's performance also depends on the size of kernels and improves the efficiency of capturing high-resolution images. Similar to contraction blocks, the bottle-neck layers also consist of the $3 \times 3$ convolution but followed by $2 \times 2$ up convolution layers. The most crucial section of Unet architecture is the expansion section. This section consists of several expansion blocks, and the number of expansion blocks should be equal to the number of contractions. Each block also contains $3 \times 3$ convolution and $2 \times 2$ up convolution layers, half of the feature maps after each block are leveraged to maintain symmetry. Moreover, the feature maps corresponding with the contraction layers also include the input. Hence, the image will be reconstructed based on the learned features while contracting the image. To produce the output, a $1 \times 1$ convolution layer is utilized to generate the feature maps with the number equal to the desired segments.

Also, *Zhou et al. (2018)*, *Zhou et al. (2019)* proposed an Unet-like architecture, namely Unet++. The advantages of Unet++ can be considered as capturing various levels of features, integrating features, and leveraging a shallow Unet structure. The discriminations between Unet and Unet++ are the skip connection associating two sub-networks and utilizing the deep supervision. The segmentation results are available at numerous nodes in the structure of Unet++ by training with deep supervision. Another Unet-like architecture

has been proposed by *Guan et al. (2020)* for detaching artifacts from 2D photoacoustic tomography images. We leverage the Unit, Unet++, and Unet2d as the decoder architectures to conduct the experiments. When compared with original Unet architecture, Unet++ and Unet2d have been built to reduce the semantic gap between the feature maps and efficiently learn the global and local features.

### Feature pyramid network

Like Unet, FPN (*Lin et al., 2017*) is also a famous architecture appropriate to segmentation tasks. FPN is a feature extractor with a single-scale image of a stochastic dimension and generates the feature maps with proportional size. The primary purpose of FPN is to build feature pyramids inside convolutional neural networks to be used in segmentation or object recognition tasks. The architecture of the FPN involves a bottom-up pathway and a top-down pathway. The bottom-up pathway defines a convolutional neural network with feature extraction, and it composes several convolution blocks. Each block consists of convolution layers. The last layer's output is leveraged as the reference set of feature maps for enriching the top-down pathway by lateral connection. Each lateral connection merges feature maps of the same spatial size from the bottom-up and top-down pathways. Thus, FPN architecture consists of a top-down pathway to construct higher resolution layers from a semantic-rich layer. To improve predicting locations' performance, we deploy the lateral to connect between reconstructed layers, and the corresponding feature maps are utilized due to the reconstructed layers are semantic strong, whereas the locations of objects are not precise. It works similarly to skip connections of ResNet.

Furthermore, the spatial resolution decreases as going up and detecting more high-level structures, the semantic value for each layer can potentially increase. Nevertheless, the bottom layers are in high resolution but can not be utilized for detection due to the semantic value is unsuitable for justifying the slow-down training computation based on it. By applying the $1 \times 1$ convolution layer at the top-down pathway, the channel dimensions of feature maps from the bottom-up pathway can be reduced and become the top-down pathway's first feature map. Furthermore, element-wise addition is applied to merge the feature maps, the bottom-up pathway, and the top-down pathway.

### Loss function for segmentation task

In deep learning and computer vision, the boundary detection definition comes from extracting features to produce significant representations of the objects. More specifically, boundary detection aims to identify object boundaries from images. Therefore, we can consider boundary detection as segmentation problems and target the boundaries to 1 and the rest of the image to 0 as the label. Thus, the loss function can be formulated with the classical function, namely Cross-entropy or Hinge loss. However, in terms of segmentation tasks, the classical loss function models can work imperfectly due to the highly unbalanced label distribution of each class and the per-pixel intrinsic of the classical loss function.

To enhance segmentation models' performance, we considered using Dice loss (*Milletari, Navab & Ahmadi, 2016*) originates from the Dice coefficient. We will introduce

more details about the Dice coefficient in 'Metrics for comparison'. The ground truth and predicted pixels can be considered as two sets. By leveraging Dice loss, the two sets are trained to overlap little by little, and the reduction of Dice loss can be obtained when the predicted pixels overlap only the ground-truth pixels. Furthermore, with Dice loss, the total number of pixels at the global scale is investigated as the denominator, whereas the numerator pays attention to the overlap between two sets at the local scale. Hence, the loss of information globally and locally is utilized by Dice loss and critically improves accuracy. Moreover, in thin boundaries, the model utilizing Dice loss can achieve better performance than others.

### Data augmentation

Deep learning requires more data to improve classification and regression tasks, although it is not easy to collect needed data. Augmentation techniques was introduced in the work of *Van Dyk & Meng (2001)*, and leveraged by *Zoph et al. (2020)*, *Frid-Adar et al. (2018)*, *Xu, Li & Zhu, 2020* and *Ahn et al. (2020)* in a vast of studies. More specific, data augmentation is the technique to abound the number of samples in the dataset by modifying the existing samples or generating newly synthetic data. Leveraging the advantages of the augmentation approach can help reduce over-fitting during the training section. In terms of image segmentation, the most general techniques for data augmentation are adjusting brightness or contrast, zoom in/out, cropping, shearing, rotation, noise, or flipping.

In this study, we adapted several techniques as follows. Firstly, the outlier will be cut off from the original image, and we set 10% and 90% for the cut-off lower and upper percentiles, respectively. Then, the low resolution of the cut-off outlier image is simulated. We also applied the mirror, contrast, and brightness transform. Furthermore, gamma and Gaussian noise are utilized before adding spatial transforms. Figure 3 visualizes the original medical image, and the image leveraged data augmentation techniques.

## EVALUATION

In this section, we present our experimental configurations in 'Settings for experiment'. Furthermore, the research information referred as the COVID-19 dataset and the considered evaluation metrics are explained in 'Dataset' and 'Metrics for comparison' respectively.

### Settings for experiment

We implemented the trained models built based on Pytorch framework (https://pytorch.org). We also accelerated the training section by utilizing the weights trained on the ImageNet dataset (*Russakovsky et al., 2015*). The processing includes three main phases: training and validation, as described in Fig. 1. For the first stage, the medical images dataset is split into training and validation sections. Afterward, we evaluated several segmentation architectures and stored internal parameters with the highest performance. A robust computational resource is required for the segmentation model. Thus, we used a server with configurations listed in Table 3 to conduct our experiments.

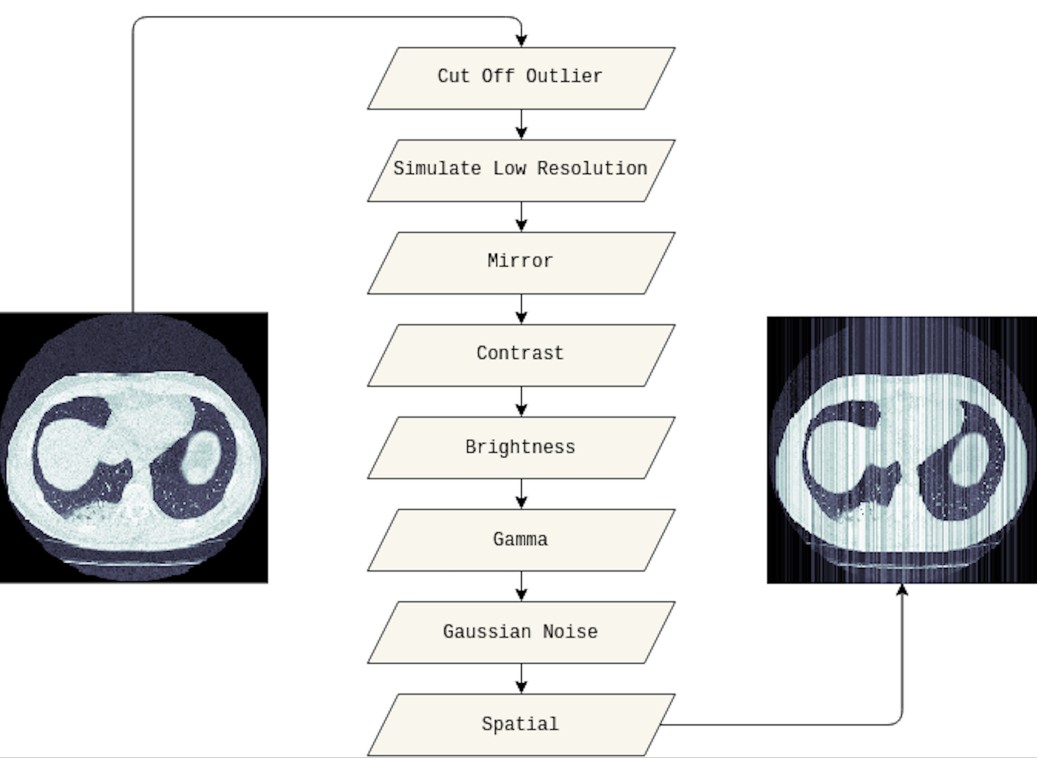

**Figure 3 The visualization of the data augmentation techniques.** The original medical image is on the left, whereas the image with data augmentation techniques is on the right.

| Name | RAM | CPU | GPU | OS |
|---|---|---|---|---|
| **Table 3 Hardware and software configurations.** | | | | |
| Description | 64 GB | Intel® i9-10900F CPU @ 2.80 GHz | NVIDIA GeForce GTX 2060 SUPER | Ubuntu 20.04 LTS |

## Dataset

We investigated our approach's performance on the COVID-19 segmentation dataset from the Italian Society of Medical and Interventional Radiology (http://medicalsegmentation.com/covid19/). The dataset includes 829 slices belonging to nine axial volumetric CTs. Furthermore, the experienced radiologist has evaluated, segmented, and marked as COVID-19 on 373 out of the total of 829 slices. The medical images have been transformed to greyscaled with $630 \times 630$ and in NIFTI file format. The segmented labels include the infection masks but also lung masks. Therefore, it could be more attractive for performing segmentation tasks on this dataset. Figure 4 visualizes a sample of the dataset; the left CT slide presents the original image of a COVID-19 patient, the right image includes the lung, and the infection region visualizing by blue and orange color, respectively.

The further information of the dataset is detailed in Table 4. The dataset includes 829 total samples (infection masks, lung masks, and images). However, the number of

CT slice of COVID-19 patient      CT slice with Lung and COVID-19 region

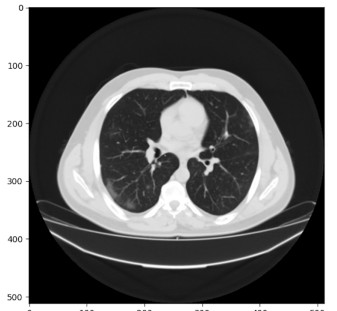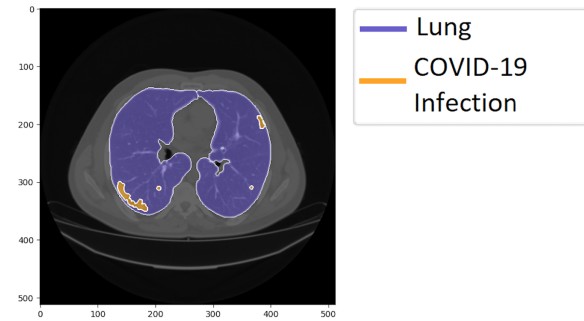

**Figure 4 A sample of the COVID-19 segmentation dataset.** The left image presents a CT slice of a COVID-19 patient, whereas the right image visualizes the lung and infection region of the patient.

**Table 4 The information of the considered dataset.**

| Type | # of samples |
| --- | --- |
| Lung Masks | 829 |
| Infection Masks | 829 |
| Infection Masks with COVID-19 | 373 |
| Training set | 300 |
| Testing set | 73 |

COVID-19 masks is 373 out of 829 samples. This work proposes COVID diagnosis research to reduce pandemics' effect so only images labeled as COVID-19 are selected and split into training and testing tests in this study. The training test consists of 300 samples, whereas the testing set includes 73 images. Furthermore, we have transformed the NIFTI files to DCM format, the lung, and infection masks into PNGs format to perform our evaluation. The images are normalized to the range of 0–255 and rescaled to a resolution in 512 × 512 pixels reducing the computation cost.

## Metrics for comparison

In this study, we considered using two metrics to evaluate the current approach's performance, namely dice (*Sørensen, 1948*; *Dice, 1945*) and Jaccard coefficients. Intuitively, the segmentation performance is measured by evaluating the overlap between the predictions and the ground-truth object. The results with more overlap regions with the ground truth reveal better performance than those with fewer overlap regions. Both Dice and Jaccard indices are in the range between 0 and 1. We assume that *A* and *B* are prediction and ground-truth masks, respectively, for a given class. If *A* and *B* match perfectly, the value of both Dice and Jaccard indices is equal to one. Otherwise, *A* and *B* are no overlap, and the value is equal to 0. Thus, the Dice coefficient can be evaluated by 2 × the area of overlap divided by the total number of pixels in both masks as in Eq. (1).

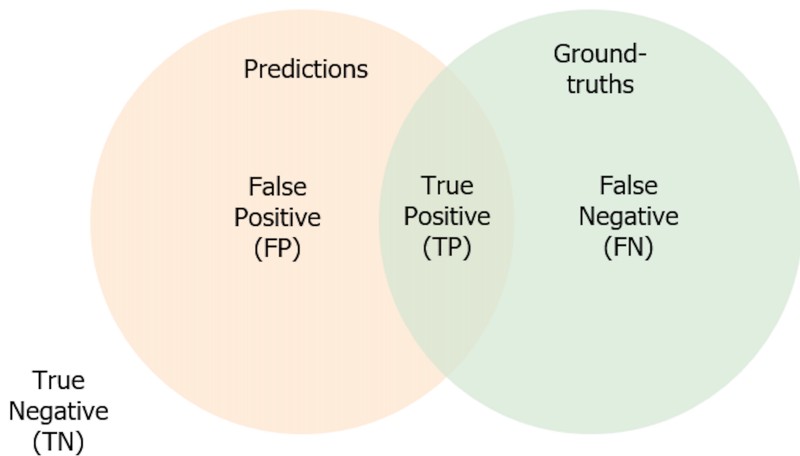

**Figure 5** **The illustration of the segmentation error.**

$$Dice(A, B) = \frac{2 \times ||A \cap B||}{||A|| + ||B||} \qquad (1)$$

Like the Dice coefficient, the Jaccard index is also one of the most common metrics in image segmentation tasks. Jaccard index can be referred to as Intersection-Over-Union (IoU). Generally, the IoU is the overlap regions between the predicted and the ground-truth mask divided by the union region between the predicted and the ground truth mask. In terms of binary or multi-class segmentation, we need to calculate the IoU of each class and average them to achieve the mean IoU. The Jaccard index or IoU can be calculated as in Eq. (2). Besides, the Precision, Recall, and $F_1$ score are also utilized to evaluate the segmentation performance.

$$Jaccard(A, B) = IoU = \frac{||A \cap B||}{||A \cup B||} \qquad (2)$$

We also consider the following definitions:

- True Positive (TP) reveals the number of positives. In other words, the predictions match with the ground-truth label.
- True Negative (TN) indicates the predictions do not belong to the ground-truth and are not segmented.
- False Positive (FP) demonstrates the predicted masks unmatch with the ground-truth masks.
- False Negative (FN) expresses the predictions belong to the ground-truth, but it is not segmented correctly.

Furthermore, in terms of the confusion matrix, the Dice and IoU equation can be rephrased as in Eqs. (3) and (4). We also present the measuring of the segmentation errors in Fig. 5.

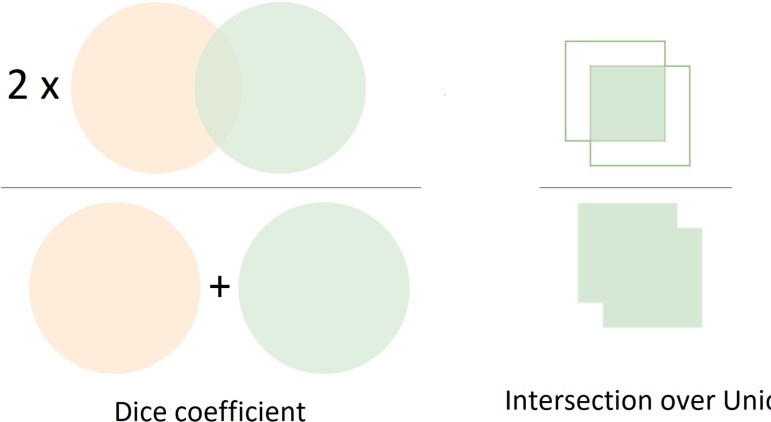

Dice coefficient          Intersection over Union

**Figure 6 The visualization of Dice and IoU.** The left image presents the Dice coefficient, the right describes the IoU.                               

$$Dice = \frac{2TP}{2TP + FP + FN} \tag{3}$$

$$Jaccard = IoU = \frac{TP}{TP + FP + FN} \tag{4}$$

The formula of computing Dice is relevant to the $F_1$ score. In other words, the Dice and $F_1$ achieve the same value in comparison with each other. Moreover, Fig. 6 visualizes in detail the differences between Dice and Jaccard/Iou indices. In Figure 6, the left image represents the Dice coefficient, whereas the right image exhibits the Intersection over Union between the predicted mask and the ground-truth mask.

## EXPERIMENTAL RESULTS

In this section, we present in detail our experimental results. 'Segmentation performance on the medical image dataset' presents the segmentation performance of the configurations introduced in Table 1. All the models are trained over 100 epochs, and the model that achieved the best performance will be stored for inferring purposes. Afterward, the discussion of the proposed methods and the other systems is presented in 'Benchmark'.

### Segmentation performance on the medical image dataset

We describe the results of both cases, non-augmented and augmented data in Tables 5 and 6, respectively. More specifically, the IoU, mean IoU (mIoU), Precision, Recall, and $F_1$ score of our experimental configurations are reported to express the performance when applying data augmentation techniques and vice versa.

Table 5 reveals the results of inferring the trained model without using data augmentation techniques. The architectures built based on the Unet decoder family obtained better results compared to FPN. Among the proposed configurations, $C_9$ acquires the best mIoU of 0.9262, whereas $C_7$ gets 0.9259 in the second place. In terms of the Dice coefficient, $C_9$ achieves 0.9016 for the COVID-19 category, being the best architecture.

**Table 5 The experimental results in details of various configurations described in Table 2.** The data augmentation techniques are not utilized to perform the experiment.

| Configuration | | $C_1$ | $C_2$ | $C_3$ | $C_4$ | $C_5$ | $C_6$ | $C_7$ | $C_8$ | $C_9$ | $C_{10}$ |
|---|---|---|---|---|---|---|---|---|---|---|---|
| IoU | Not Lung Nor COVID-19 | 0.9954 | 0.9948 | 0.9956 | 0.9956 | 0.9941 | 0.9956 | 0.9968 | 0.9956 | 0.9971 | 0.9954 |
| | Lung | 0.9449 | 0.9407 | 0.9393 | 0.9464 | 0.9315 | 0.9492 | 0.9575 | 0.9475 | 0.9581 | 0.9460 |
| | COVID-19 | 0.7978 | 0.7860 | 0.7631 | 0.8059 | 0.7498 | 0.8175 | 0.8233 | 0.8177 | 0.8233 | 0.8127 |
| **mIOU** | | 0.9127 | 0.9072 | 0.8993 | 0.9160 | 0.8918 | 0.9208 | 0.9259 | 0.9204 | 0.9262 | 0.9219 |
| $F_1$-score/Dice | Not Lung Nor COVID-19 | 0.9977 | 0.9975 | 0.9978 | 0.9978 | 0.9971 | 0.9978 | 0.9975 | 0.9977 | 0.9977 | 0.9978 |
| | Lung | 0.9717 | 0.9695 | 0.9687 | 0.9725 | 0.9645 | 0.9724 | 0.9716 | 0.9725 | 0.9735 | 0.9722 |
| | COVID-19 | 0.8875 | 0.8802 | 0.8656 | 0.8925 | 0.8570 | 0.8996 | 0.8934 | 0.8993 | 0.9016 | 0.8967 |
| Precision | Not Lung nor COVID-19 | 0.9986 | 0.9982 | 0.9981 | 0.9986 | 0.9988 | 0.9974 | 0.9971 | 0.9981 | 0.9977 | 0.9981 |
| | Lung | 0.9691 | 0.9721 | 0.9656 | 0.9747 | 0.9633 | 0.9764 | 0.9707 | 0.9706 | 0.9718 | 0.9749 |
| | COVID-19 | 0.8683 | 0.8386 | 0.8737 | 0.8502 | 0.8036 | 0.8883 | 0.9145 | 0.8979 | 0.9129 | 0.8730 |
| Recall | Not Lung nor COVID-19 | 0.9968 | 0.9967 | 0.9975 | 0.9969 | 0.9953 | 0.9981 | 0.9979 | 0.9974 | 0.9977 | 0.9976 |
| | Lung | 0.9742 | 0.9669 | 0.9718 | 0.9702 | 0.9658 | 0.9684 | 0.9725 | 0.9745 | 0.9752 | 0.9696 |
| | COVID-19 | 0.9077 | 0.9261 | 0.8577 | 0.9394 | 0.9180 | 0.9112 | 0.8732 | 0.9008 | 0.8906 | 0.9233 |
| **Configuration** | | $C_{11}$ | $C_{12}$ | $C_{13}$ | $C_{14}$ | $C_{15}$ | $C_{16}$ | $C_{17}$ | $C_{18}$ | $C_{19}$ | $C_{20}$ |
| IoU | Not Lung nor COVID-19 | 0.9959 | 0.9956 | 0.9959 | 0.9956 | 0.9950 | 0.9941 | 0.9942 | 0.9948 | 0.9944 | 0.9933 |
| | Lung | 0.9488 | 0.9476 | 0.9475 | 0.9415 | 0.9358 | 0.9351 | 0.9305 | 0.9379 | 0.9369 | 0.9234 |
| | COVID-19 | 0.8168 | 0.8138 | 0.8073 | 0.7776 | 0.7608 | 0.7745 | 0.7311 | 0.7758 | 0.7786 | 0.7299 |
| mIOU | | 0.9205 | 0.9191 | 0.9169 | 0.9049 | 0.8972 | 0.9013 | 0.8852 | 0.9029 | 0.9033 | 0.8822 |
| $F_1$-score/Dice | Not Lung nor COVID-19 | 0.9979 | 0.9978 | 0.9979 | 0.9978 | 0.9975 | 0.9971 | 0.9971 | 0.9974 | 0.9972 | 0.9966 |
| | Lung | 0.9737 | 0.9731 | 0.9731 | 0.9646 | 0.9668 | 0.9665 | 0.9640 | 0.9680 | 0.9674 | 0.9602 |
| | COVID-19 | 0.8992 | 0.8973 | 0.8934 | 0.8749 | 0.8641 | 0.8729 | 0.8446 | 0.8738 | 0.8755 | 0.8439 |
| Precision | Not Lung nor COVID-19 | 0.9988 | 0.9991 | 0.9989 | 0.9975 | 0.9981 | 0.9987 | 0.9986 | 0.9986 | 0.9982 | 0.9978 |
| | Lung | 0.9726 | 0.9689 | 0.9744 | 0.9752 | 0.9722 | 0.9625 | 0.9666 | 0.9709 | 0.9665 | 0.9647 |
| | COVID-19 | 0.8742 | 0.8711 | 0.8522 | 0.8548 | 0.8139 | 0.8350 | 0.7823 | 0.8187 | 0.8445 | 0.7822 |
| Recall | Not Lung nor COVID-19 | 0.9971 | 0.9964 | 0.9970 | 0.9981 | 0.9969 | 0.9954 | 0.9955 | 0.9962 | 0.9962 | 0.9954 |
| | Lung | 0.9749 | 0.9773 | 0.9717 | 0.9646 | 0.9615 | 0.9705 | 0.9615 | 0.9651 | 0.9684 | 0.9557 |
| | COVID-19 | 0.9254 | 0.9252 | 0.9386 | 0.8959 | 0.9209 | 0.9144 | 0.9177 | 0.9370 | 0.9089 | 0.9162 |

Considering Precision and Recall, $C_9$ earns 0.9129, 0.8906, and gives a comparable performance to the others. Table 6 presents the performance of our experimental configurations. The proposed networks are trained under data augmentation techniques. Overall, almost the configuration with decoder-based Unet family ($C_1$ to $C_{15}$) achieved mIoU over 0.9, whereas the architecture-based FPN obtained approximately at 0.8. Between the configurations, $C_6$ achieves the best performance with the mIoU obtained of 0.9283. Configuration $C_6$ represents the combination of ResNet and Unet2d. Furthermore, the second place gets the mIoU of 0.9234 with configuration $C_{13}$ trained with SE ResNeXt and Unet++ model as the encoder and decoder.

As for lung and COVID-19 segmentation, the average IoU achieves 0.8 on overall configurations for COVID-19 infection regions and 0.94 for lung masks. The configurations of $C_6$ and $C_{13}$ exhibit promising performance on the COVID-19 segmentation task with obtained IoU of 0.8241 and 0.8234. Meanwhile, the configurations

**Table 6 Experimental results in details with various configurations of the architectures mentioned and described in Table 2.**

| Configuration | | $C_1$ | $C_2$ | $C_3$ | $C_4$ | $C_5$ | $C_6$ | $C_7$ | $C_8$ | $C_9$ | $C_{10}$ |
|---|---|---|---|---|---|---|---|---|---|---|---|
| IoU | Not Lung Nor COVID-19 | 0.9956 | 0.9951 | 0.9952 | 0.9958 | 0.9945 | 0.9981 | 0.9952 | 0.9951 | 0.9955 | 0.9953 |
| | Lung | 0.9468 | 0.9422 | 0.9351 | 0.9477 | 0.9341 | 0.9628 | 0.9443 | 0.9492 | 0.9462 | 0.9465 |
| | COVID-19 | 0.8121 | 0.7868 | 0.7569 | 0.8103 | 0.7581 | 0.8241 | 0.8169 | 0.8193 | 0.8147 | 0.8239 |
| **mIOU** | | 0.9182 | 0.9081 | 0.8957 | 0.9181 | 0.8955 | 0.9283 | 0.9188 | 0.9212 | 0.9188 | 0.9219 |
| $F_1$-score/Dice | Not Lung Nor COVID-19 | 0.9978 | 0.9976 | 0.9976 | 0.9979 | 0.9972 | 0.9978 | 0.9976 | 0.9979 | 0.9976 | 0.9977 |
| | Lung | 0.9727 | 0.9703 | 0.9664 | 0.9731 | 0.9661 | 0.9734 | 0.9714 | 0.9737 | 0.9714 | 0.9725 |
| | COVID-19 | 0.8963 | 0.8807 | 0.8616 | 0.8952 | 0.8623 | 0.8941 | 0.8992 | 0.9003 | 0.8979 | 0.9034 |
| Precision | Not Lung nor COVID-19 | 0.9985 | 0.9986 | 0.9983 | 0.9985 | 0.9981 | 0.9972 | 0.9977 | 0.9981 | 0.9978 | 0.9974 |
| | Lung | 0.9721 | 0.9712 | 0.9651 | 0.9735 | 0.9712 | 0.9761 | 0.9726 | 0.9732 | 0.9697 | 0.9776 |
| | COVID-19 | 0.8716 | 0.8393 | 0.8432 | 0.8709 | 0.8061 | 0.9025 | 0.8884 | 0.8948 | 0.9027 | 0.8844 |
| Recall | Not Lung nor COVID-19 | 0.9971 | 0.9965 | 0.9969 | 0.9973 | 0.9964 | 0.9984 | 0.9975 | 0.9976 | 0.9975 | 0.9979 |
| | Lung | 0.9733 | 0.9694 | 0.9677 | 0.9728 | 0.9607 | 0.9707 | 0.9702 | 0.9743 | 0.9732 | 0.9675 |
| | COVID-19 | 0.9224 | 0.9263 | 0.8801 | 0.9209 | 0.9272 | 0.9258 | 0.9103 | 0.9059 | 0.8932 | 0.9233 |
| Configuration | | $C_{11}$ | $C_{12}$ | $C_{13}$ | $C_{14}$ | $C_{15}$ | $C_{16}$ | $C_{17}$ | $C_{18}$ | $C_{19}$ | $C_{20}$ |
| IoU | Not Lung nor COVID-19 | 0.9958 | 0.9956 | 0.9961 | 0.9954 | 0.9936 | 0.9947 | 0.9941 | 0.9951 | 0.9945 | 0.9931 |
| | Lung | 0.9498 | 0.9476 | 0.9508 | 0.9465 | 0.9293 | 0.9327 | 0.9341 | 0.9308 | 0.9344 | 0.9235 |
| | COVID-19 | 0.8186 | 0.8138 | 0.8234 | 0.8232 | 0.7671 | 0.7495 | 0.7616 | 0.7395 | 0.7524 | 0.7356 |
| mIOU | | 0.9214 | 0.9191 | 0.9234 | 0.9217 | 0.8966 | 0.8923 | 0.8966 | 0.8884 | 0.8938 | 0.8841 |
| $F_1$-score/Dice | Not Lung nor COVID-19 | 0.9979 | 0.9978 | 0.9981 | 0.9977 | 0.9968 | 0.9973 | 0.9971 | 0.9974 | 0.9973 | 0.9965 |
| | Lung | 0.9742 | 0.9731 | 0.9748 | 0.9725 | 0.9633 | 0.9652 | 0.9659 | 0.9642 | 0.9661 | 0.9602 |
| | COVID-19 | 0.9002 | 0.8973 | 0.9032 | 0.9031 | 0.8681 | 0.8568 | 0.8647 | 0.8502 | 0.8587 | 0.8476 |
| Precision | Not Lung nor COVID-19 | 0.9989 | 0.9991 | 0.9989 | 0.9991 | 0.9986 | 0.9975 | 0.9981 | 0.9975 | 0.9965 | 0.9982 |
| | Lung | 0.9721 | 0.9689 | 0.9741 | 0.9669 | 0.9583 | 0.9726 | 0.9682 | 0.9737 | 0.9721 | 0.9613 |
| | COVID-19 | 0.8762 | 0.8711 | 0.8735 | 0.8836 | 0.8327 | 0.8102 | 0.8172 | 0.7988 | 0.8528 | 0.7874 |
| Recall | Not Lung nor COVID-19 | 0.9969 | 0.9964 | 0.9971 | 0.9963 | 0.9945 | 0.9971 | 0.9959 | 0.9974 | 0.9981 | 0.9948 |
| | Lung | 0.9765 | 0.9773 | 0.9755 | 0.9781 | 0.9684 | 0.9579 | 0.9637 | 0.9548 | 0.9601 | 0.9591 |
| | COVID-19 | 0.9256 | 0.9252 | 0.9349 | 0.9234 | 0.9066 | 0.9091 | 0.9179 | 0.9088 | 0.8648 | 0.8879 |

of $C_{13}$ and $C_{14}$ get the maximum $F_1$ score for the COVID-19 category with an obtained value of 0.9032 and 0.9031. By examining Precision and Recall, $C_6$ acquires the best precision performance, whereas $C_{13}$ gains the maximum Recall. Figure 7 depicts the confusion matrix for configuration $C_{13}$. We also normalize the confusion matrix over rows for analyzing purposes. The confusion matrix values reveal that most of the misjudgments of COVID-19 infection regions are categorized as lung and vice versa.

The comparison of not using data augmentation and using data augmentation reveals the architectures with FPN-based tend to be more effective when not utilizing augmentation techniques. It can be demonstrated by the results of $C_{16}$–$C_{18}$. By the Unet family decoder, the outcomes depict that the models with nearly equivalent performance, $i.e.$, $C_2$, $C_4$, $C_8$, $C_{10}$, and $C_{12}$. The most discriminate configurations are $C_{11}$ and $C_{14}$. The performance of $C_{11}$ and $C_{14}$ are strongly boosted by applying data augmentation techniques. Furthermore, $C_{11}$ and $C_{14}$ also obtain better performance. In

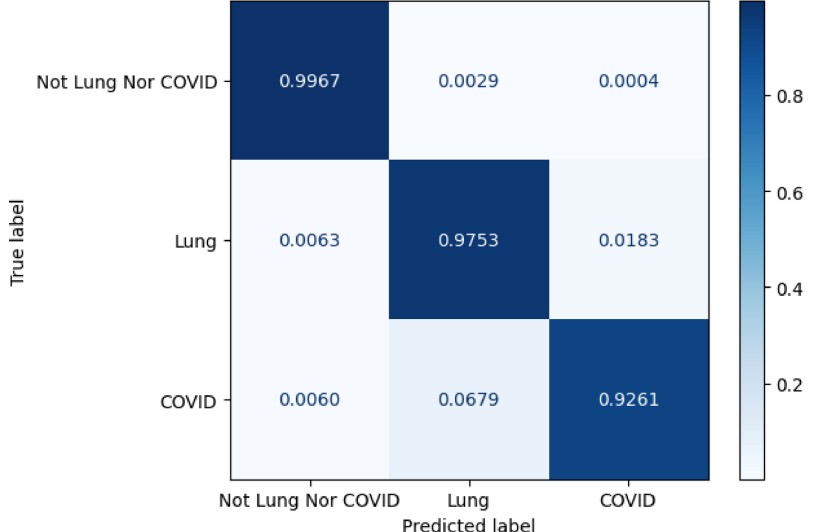

**Figure 7 The visualization of confusion matrix for segmentation model-based SE ResNeXt and Unet++.**

**Table 7 The report (in second(s)) of training time (1) and inference time (2) of 20 configurations.**

|     | $C_1$ | $C_2$ | $C_3$ | $C_4$ | $C_5$ | $C_6$ | $C_7$ | $C_8$ | $C_9$ | $C_{10}$ |
|-----|-------|-------|-------|-------|-------|-------|-------|-------|-------|----------|
| (1) | 5,949 | 6,390 | 7,900 | 6,659 | 6,248 | 7,351 | 7,377 | 7,497 | 7,465 | 7,526 |
| (2) | 0.5753 | 0.5892 | 0.6164 | 0.6027 | 0.6301 | 0.5891 | 0.6438 | 0.6287 | 0.6287 | 0.6301 |
|     | $C_{11}$ | $C_{12}$ | $C_{13}$ | $C_{14}$ | $C_{15}$ | $C_{16}$ | $C_{17}$ | $C_{18}$ | $C_{19}$ | $C_{20}$ |
| (1) | 6,656 | 8,884 | 10,207 | 9,147 | 6,505 | 5,832 | 6,127 | 7,481 | 6,388 | 5,999 |
| (2) | 0.6109 | 0.6931 | 0.7328 | 0.7027 | 0.5986 | 0.5684 | 0.5889 | 0.6054 | 0.5972 | 0.5794 |

this respect, we can conclude that the augmentation techniques affect the results with almost all configurations.

Furthermore, the training and inference times are reported in Table 7. The training presents the total time needs for 100 epochs, whereas the inference expresses the average time to segment each slice. With the Unet and FPN decoder, the architectures are trained with lower computation costs than the rest. The least and most time-consuming configurations for training/inference are $C_{16}$ and $C_{13}$, respectively. As observed from the results, with the same encoders, architectures FPN decoder-based segment fastest for each slice.

We also visualize several samples with ground truth and the prediction masks in and Fig. 8. The lung is colored by slate blue and orange for COVID-19. As observed from the results, our predictions express promising segmentation performance. In terms of complex COVID-19 infection regions as in Figs. 8B, 8C, or 8E, the boundary of COVID-19 segmentation is equivalent to the corresponding ground-truth. For the different view as in Figs. 8A and 8D, the interesting regions are segmented quite correctly. Meanwhile, the lungs are also produced identically in comparison with the ground truth.

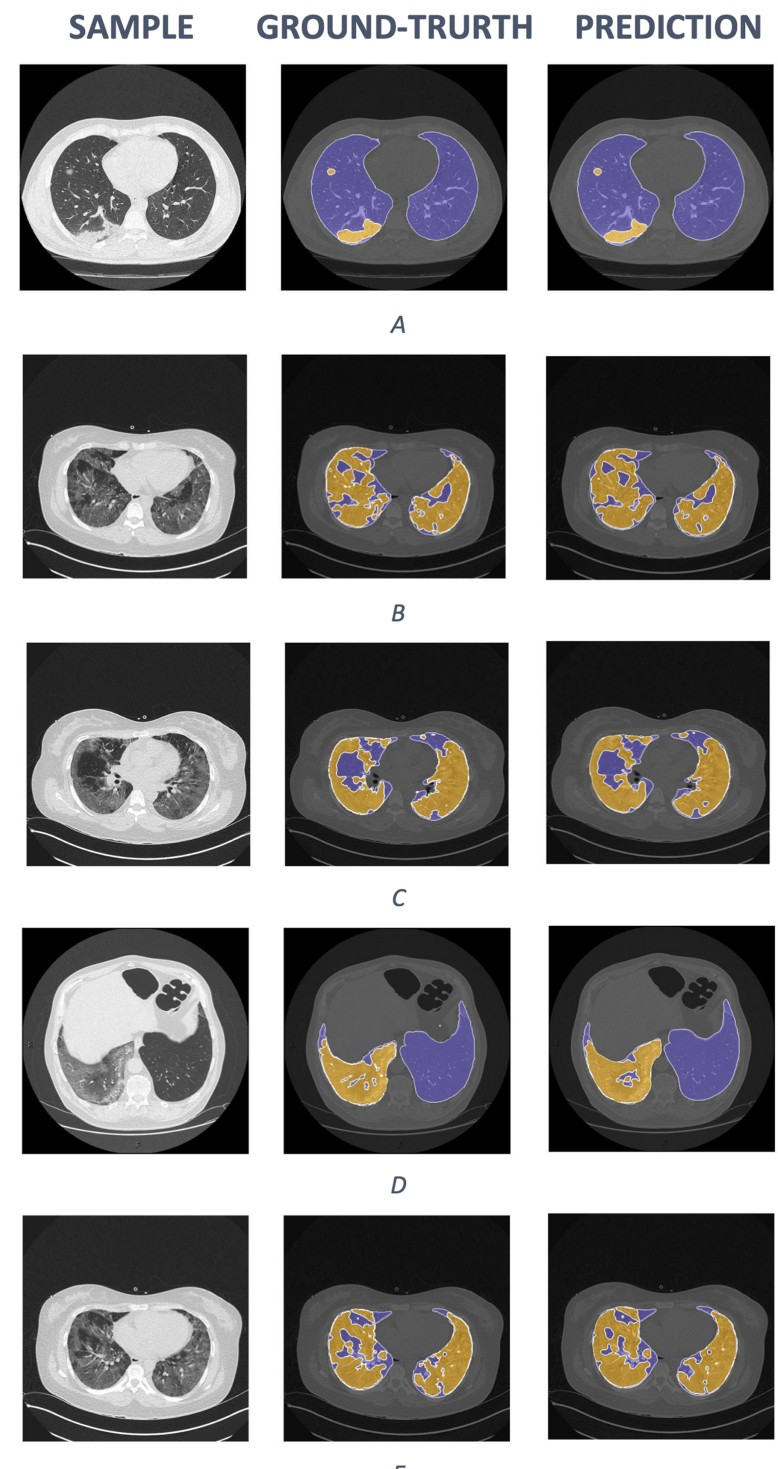

**SAMPLE   GROUND-TRURTH   PREDICTION**

A

B

C

D

E

**Figure 8 The COVID-19 samples, ground-truth and predictions from test set.** The purple regions denote the lung, whereas the yellows represent the COVID-19 infection.

**Table 8 The performance of lung segmentation, the lung includes non of infected COVID-19 region.**

|  | Method | Dice | Sensitivity |
|---|---|---|---|
| Our configurations | C4 | 0.9731 | 0.9728 |
|  | C6 | 0.9734 | 0.9707 |
|  | C13 | 0.9748 | 0.9755 |
|  | C19 | 0.9661 | 0.9601 |
| *Saood & Hatem (2021)* | SegNet | 0.7490 | 0.9560 |

**Table 9 The summarization of quantitative results of infected COVID-19 regions.** A '–' symbol means that there was no relevant information in the original study.

|  | Method | Dice | Sensitivity | Precision |
|---|---|---|---|---|
| Our configurations | C1 | 0.8963 | 0.9224 | 0.8716 |
|  | C10 | 0.9034 | 0.9233 | 0.8844 |
|  | C13 | 0.9032 | 0.9349 | 0.8711 |
|  | C17 | 0.8966 | 0.9179 | 0.8172 |
| Umit *Budak et al. (2021)* | A-SegNet + FTL | 0.8961 | 0.9273 | – |
| *Zhou, Canu & Ruan (2020)* | Unet + attention mechanism | 0.8310 | 0.8670 | – |
| *Raj et al. (2021)* | ADID-UNET | 0.8031 sw | 0.7973 | 0.8476 |

## Benchmark

### Lung segmentation

The comparison of lung segmentation is presented in Table 8. As observed from the results, our configurations with different decoders have outperformed the approach by *Saood & Hatem (2021)*. More specifically, the work of *Saood & Hatem (2021)* obtained a Dice of 0.7490 and Sensitivity of 0.9560 with the SegNet method, while our best configuration, $C_{13}$, achieved a maximum Dice of 0.9748 and Sensitivity of 0.9755. In particular, by leveraging SE ResNeXt and Unet++ architecture, we get the maximum score compared to the others.

### COVID-19 segmentation

Table 9 reveals the the compared performance of the proposed approach with the state-of-the-art methods including SegNet (*Budak et al., 2021*), Unet (*Zhou, Canu & Ruan, 2020*) with attention mechanism, and ADID-UNET (*Raj et al., 2021*). We compared our configurations with the others on Dice, Sensitivity, and Precision values. Our approach achieves a better performance in terms of Dice, Sensitivity, and Precision concerning three other methods. Specifically, the study of *Budak et al. (2021)* acquired a Dice of 0.8961 and a Sensitivity of 0.9273 with SegNet. Also, *Zhou, Canu & Ruan (2020)* leveraged Unet with attention mechanism and obtained the performance of 0.8310 and 0.8670 by Dice and Sensitivity, respectively. The method of *Raj et al. (2021)* achieved a Dice of 0.8031, Sensitivity of 0.7973, and Precision of 0.8476 with ADID-UNET. In particular, using configuration $C_{10}$, the segmentation performance obtained the best performance with 0.9034 in Dice, Sensitivity of 0.9233, and Precision of 0.8844, whereas $C_{13}$ achieved a

Dice of 0.9032, Sensitivity of 0.9349, and Precision of 0.8711. Table 9, the architecture based on Unet++ decoder is the best segmentation model among the others. Our configurations get a promising performance compared with the others.

## CONCLUSION

In this study, we systematically presented a viable solution for lung and COVID-19 segmentation from CT images. The proposed model is implemented based on the convolutional neural networks, _i.e._, ResNet, ResNeSt, SE ResNeXt, Res2Net, or EfficientNet decoder-based Unet family and Feature Pyramid Network as the encoders and decoders of segmentation models. We evaluated the proposed method by the open segmentation dataset with numerous model structures. The experimental results reveal that the model with the decoder-based Unet family obtained better performance than FPN. Furthermore, the segmentation results are compared with the ground truth annotated by an experienced radiologist and exhibit promising performance. More specifically, the best architecture obtained a mIoU of 0.9234, 0.9032 of $F_1$-score, 0.8735, and 0.9349 of Precision and Recall, respectively. Also, segmenting the minimal infection regions still challenges us due to their size and ambiguous regions.

### Funding

Toan Bao Tran was funded by the Vingroup Joint Stock Company and supported by the Domestic Master/Ph.D. Scholarship Programme of the Vingroup Innovation Foundation (VINIF), Vingroup Big Data Institute (VINBIGDATA), code VINIF.2020.ThS63. The funders had no role in study design, data collection and analysis, decision to publish, or preparation of the manuscript.

### Grant Disclosures

The following grant information was disclosed by the authors:
Vingroup Joint Stock Company.
Domestic Master/Ph.D. Scholarship Programme of Vingroup Innovation Foundation (VINIF).
Vingroup Big Data Institute (VINBIGDATA): VINIF.2020.ThS63.

### Competing Interests

The authors declare that they have no competing interests.

### Author Contributions

- Hai Thanh Nguyen conceived and designed the experiments, performed the experiments, analyzed the data, performed the computation work, prepared figures and/or tables, authored or reviewed drafts of the paper, and approved the final draft.
- Toan Bao Tran conceived and designed the experiments, performed the experiments, analyzed the data, performed the computation work, prepared figures and/or tables, authored or reviewed drafts of the paper, and approved the final draft.

- Huong Hoang Luong performed the experiments, analyzed the data, prepared figures and/or tables, authored or reviewed drafts of the paper, and approved the final draft.
- Tuan Khoi Nguyen Huynh analyzed the data, performed the computation work, prepared figures and/or tables, and approved the final draft.

## Data Availability

This study's experimental scripts and code are available at GitHub: https://github.com/nthai-cit/covid-seg.

The data that support the findings of this study are available at https://medicalsegmentation.com/covid19/.

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
