# Peer review of "Decoders configurations based on Unet family and feature pyramid network for COVID-19 segmentation on CT images"

_PeerJ Computer Science, doi:10.7717/peerj-cs.719_

## Round 0.1 · original submission · Major Revisions

The paper has merits. Please revise your paper based on the comments.

Reviewer 1 ·

Basic reporting

English needs to be moderately revised to be Grammarly correct.
What is the novelty of the proposed research, what is your main contribution?
Could you please explain the difference the Unet family?

“Ronneberger et al. (2015a) and ? demonstrated that the improvement of segmentation tasks recently relied 138 more on the encoder-decoder than other architecture.” What does the question mark mean?
Do you mean that all the features from Unet family are used? Or any specific Unet is finally used?
What the meaning of Cn?
Check equation 2 and equation 7? Why do you have two definitions for Jaccard?
Could you compare your proposed method with the recently published method, such as “Covid-19 Classification by FGCNet with Deep Feature Fusion from Graph Convolutional Network and Convolutional Neural Network”
“Diagnosis of COVID-19 by Wavelet Renyi Entropy and Three-Segment Biogeography-based Optimization”.

Experimental design

See the basic reporting

Validity of the findings

See the basic reporting

Reviewer 2 ·

Basic reporting

For the most part, this report is clear and well organized. I very much appreciate the clear structure of the paper and the concise language. However, there are still some minor issues throughout the paper.
Here are several issues that I found.
1. In the abstract, line 24 to 26 has grammar issues.
2. There are almost no citations that are included in the introduction part.
3. Line 137 to 140 has some grammar issues.
4. Quality of Figure 6 can be improved
5. Table 5 might be more clear if it's rotated by 90 degrees.
6. Equation 2 is weirdly placed.
7. The discussion about recall precision and F1 score is a bit redundant since it is a fairly commonly used metric.

Experimental design

The experiment of this paper is well designed. However, It might be beneficial to include the result of the segmentation on the unaugmented image also. Given that in the benchmark papers, the authors didn't use the same augmentation step, it is a weaker claim to say that the improvement in results presented in this work mainly comes from the superior structure without showing those results.

Validity of the findings

The finding of the paper is valid.

Reviewer 3 ·

Basic reporting

English writing is not clear, many sentences are awkward.
In many places upper-case letters are used instead of lower case where is not necessary.
The text in most figures is not clear. Try saving your drawings as tiff images and not pngs. Make them bigger from the beginning.
The introduction is totally unrelated to the paper’s topic. The paper is about developing, identifying, and evaluating deep learning methods for image segmentation and not about information technology in medicine.
The related works section is very badly written. It is just an enumeration of citations with no descriptions in between. It doesn’t tell a story nor it presents a clear logical
Lines 115-119 in the Methods section, should be moved at the end of the Introduction.

Line 42 – “We” needs to be changes to lowercase “we”.
Line 58 - “Imaging” needs to be changes to lowercase “imaging”.
Line 137 – There is a missing reference.
Figure 6 and 9 – Text is not clear. Try to improve the quality of your image.

Experimental design

This paper proposes a new improved method that combines pretrained ResNet architectures with Unet++ for image segmentation. The application described here is lung segmentation in CT images.
The research question addressed in the paper is unclear.
The methods section is more like a second related works section. Evaluation metrics are described in more details compared to the actual methods.

Validity of the findings

It looks like the proposed approach improves the results compared to other methods, but at what cost. No training or inference time are reported in the paper. What is the main advantage of the presented method?
The paper specifies a github repository, however the code uploaded there is not enough to replicate the experiments in the paper.
It is unclear to me what makes some of the 20 configurations new.

Additional comments

The proposed method looks more like a brute force approach than a new improved method.
The paragraph about a graphical interface is irrelevant to the paper.

---

## Round 0.2 · Minor Revisions

The article is close to Acceptance. However, we noticed that you chose to include the references suggested by Reviewer 1. After a review, PeerJ staff and I do not feel that these references were the most appropriate to have included. Therefore, I ask you to critically evaluate the citations which were added and remove them, unless you are strongly of the opinion that they are highly relevant to your manuscript.

Reviewer 1 ·

Basic reporting

NONE

Experimental design

NONE

Validity of the findings

NONE

Additional comments

None

Reviewer 2 ·

Basic reporting

The paper meets all criteria in basic reporting.

Experimental design

The experimental design is solid.

Validity of the findings

The findings in the paper are valid

---

## Round 0.3 · accepted · Accept

I am happy to inform you that your submission has been accepted for publication.